# Endoplasmic Reticulum Stress in Gliomas: Exploiting a Dual-Effect Dysfunction through Chemical Pharmaceutical Compounds and Natural Derivatives for Therapeutical Uses

**DOI:** 10.3390/ijms25074078

**Published:** 2024-04-06

**Authors:** Daniel García-López, Montserrat Zaragoza-Ojeda, Pilar Eguía-Aguilar, Francisco Arenas-Huertero

**Affiliations:** 1Laboratorio de Investigación en Patología Experimental, Hospital Infantil de México Federico Gómez, Mexico City 06720, Mexico; dgarcia.lo1833@gmail.com (D.G.-L.); montse_zaragoza@hotmail.com (M.Z.-O.); meguia@himfg.edu.mx (P.E.-A.); 2Facultad de Ciencia y Tecnología, Universidad Simón Bolívar, Mexico City 03920, Mexico; 3Departamento de Patología Clínica y Experimental, Hospital Infantil de México Federico Gómez, Mexico City 06720, Mexico; 4Centro de Investigación en Biomedicina y Bioseguridad, Hospital Infantil de México Federico Gómez, Mexico City 06720, Mexico

**Keywords:** glioma, astrocytoma, glioblastoma, ATF6, IRE-1, PERK, endoplasmic reticulum stress

## Abstract

The endoplasmic reticulum maintains proteostasis, which can be disrupted by oxidative stress, nutrient deprivation, hypoxia, lack of ATP, and toxicity caused by xenobiotic compounds, all of which can result in the accumulation of misfolded proteins. These stressors activate the unfolded protein response (UPR), which aims to restore proteostasis and avoid cell death. However, endoplasmic response-associated degradation (ERAD) is sometimes triggered to degrade the misfolded and unassembled proteins instead. If stress persists, cells activate three sensors: PERK, IRE-1, and ATF6. Glioma cells can use these sensors to remain unresponsive to chemotherapeutic treatments. In such cases, the activation of ATF4 via PERK and some proteins via IRE-1 can promote several types of cell death. The search for new antitumor compounds that can successfully and directly induce an endoplasmic reticulum stress response ranges from ligands to oxygen-dependent metabolic pathways in the cell capable of activating cell death pathways. Herein, we discuss the importance of the ER stress mechanism in glioma and likely therapeutic targets within the UPR pathway, as well as chemicals, pharmaceutical compounds, and natural derivatives of potential use against gliomas.

## 1. Introduction

Gliomas are the most common tumors of the central nervous system (CNS) in children and adults. They are solid tumors with unique characteristics, which have represented a challenge in the study of several aspects, such as tumor biology, genomic and molecular characteristics, and mechanisms of cell survival and chemoresistance to several antineoplastic compounds. However, some cellular characteristics, such as cell survival and chemoresistance, can be used to better understand some common mechanisms and use them to trigger cell death [1]. Among the cellular responses with potential clinical applications, those of one of the organelles, the endoplasmic reticulum (ER), hold promise. ER control of protein synthesis is a phenotypic response to several types of stress that can be used in cell control and represents an essential tool in clinical oncology [2]. This review summarizes the current literature on ER responses to stress conditions and research on the interfering with this mechanism to develop new compounds (both synthetic and natural) that, alone or combined with temozolomide (TMZ), can lead to achievements such as inducing cell death, as well as improving the clinical management and control of tumors in patients.

## 2. General Characteristics and Classification of Gliomas

Gliomas are neoplasms of the CNS that can develop in the brain and spinal cord. Histologically, they comprise tumoral glial cells derived from astrocytes, oligodendrocytes, and ependymal cells. Gliomas are the most malignant brain tumors in adults; they also tend to present a high degree of intratumor genomic heterogeneity, which complicates treatment. The prognosis for patients who suffer from this kind of tumor indicates a short life expectancy of just 14 months in the most severe cases. Although gliomas have been studied for years, critical unresolved questions related to their etiology, cellular origin, cell survival, and prospects regarding their therapy remain [1]. In the 2016 WHO classification of CNS tumors, significant modifications were added when mutations in the gene encoding isocitrate dehydrogenase (IDH) and alterations in Histone 3 were recognized as characteristic in diagnoses; therefore, these were adopted as decisive markers for classifying gliomas [3,4].

In 2021, the WHO presented a reclassification of tumors of the CNS that emphasized the need to consider, at diagnosis, not only histological characteristics—the standard practice up to that time—but also molecular features, as this improves the classification of some cases and enhances the clinical focus to improve the treatments available to patients [5]. This classification highlights the molecular and genetic characteristics of adult and pediatric gliomas, as well as the location of the tumor. The classification establishes the following groups: (1) diffuse adult-type glioma, which most frequently includes IDH-mutant astrocytoma, IDH-wild-type glioblastoma, and IDH-mutant and 1p/19q-deleted oligodendroglioma; (2) diffuse pediatric low-grade gliomas, including diffuse astrocytoma altered in *MYB* or *MYBL* and diffuse low-grade glioma altered in the MAPK pathway; (3) diffuse pediatric high-grade gliomas, including midline gliomas altered in H3K27, hemispheric gliomas mutated in H3 G34, and H3 wild-type and IDH-mutant gliomas; and finally, (4) circumscribed astrocytic gliomas, with the most frequent being pilocytic astrocytoma [5]. In this new classification, malignant gliomas in adults are characterized by the IDH mutation, and those of the pediatric type are characterized by the alteration in H3K27. Low-grade gliomas in children rarely transform into high-grade tumors, unlike adult gliomas, in which the presence of the *IDH1* and *IDH2* mutations may influence the development of high-grade gliomas [6]. Diffuse gliomas are the most common type, especially in adults, and are characterized by alterations in the IDH gene; they include three types: IDH-mutant astrocytoma, IDH-mutant and 1p/19q-deleted oligodendroglioma, and IDH-wild-type glioblastoma. Studies have further detected that the histological aspect of these tumors changes entirely with IDH mutations; therefore IDH mutations have been adopted as a decisive marker for classifying gliomas [3,4]. Mutations in the active site of the *IDH1* gene generate a new enzymatic function that gives this gene the ability to convert α-ketoglutarate into D-2-hydroxyglutarate (D-2-HG), which functions as an oncometabolite. D-2-HG suppresses the function of several demethylases, causing DNA and histone methylation levels to increase, which leads to aberrant expression of oncogenic pathways and affects tumor suppressors. The degree of methylation and specific methylation sites (promoters, gene body) define two CpG island methylation phenotypes (G-CIMP): G-CIMP-high and G-CIMP-low. High- and low-grade gliomas with high IDHmut/CIMP show a better prognosis than IDHmut/low CIMP. The mechanism associated with a favorable prognosis remains unclear but may be in part due to the silencing of mesenchymal genes [7,8]. Paradoxically, patients with mutations in *IDH1* have a better prognosis than those who do not have them. This may be partially explained by the fact that D-2-HG alters DNA repair pathways, making IDH1-mutated gliomas more sensitive to radiotherapy and genotoxic agents than those without mutations [8]. An interesting point of susceptibility involves patients that present L-2-hydroxy glutaric aciduria, a rare neurometabolic disorder caused by a mutation in the L-2-hydroxyglutarate dehydrogenase gene that generates a stereoisomer of the oncometabolite D-2-HG [9,10].

Despite these significant advances in the classification of pediatric and adult tumors, patient survival has not changed in the past 20 years, a fact that underscores the need to find new forms of intervention and to better understand the underlying biology of these tumors or of the alterations in their various components, such as the ER, because this will open a window of opportunity for applications that will impact tumor control. After summarizing the ER’s general characteristics, we review its stress responses and the molecular strategies that glioma cells use to achieve survival. Based on this information, the aim is to take advantage of ER responses to develop applications using new compounds (synthetic and natural) that, alone or combined with temozolomide (TMZ), can lead to achievements such as inducing cell death, as well as improving the clinical management and control of tumors in patients.

## 3. General Characteristics of the Stress Response of the ER

### The Function of the ER

The ER has a tubular structure that forms the cisterns in which protein synthesis takes place and is believed to form almost one-third of all the proteins that the cell requires [2]. The ER varies in each cell type because marked differences exist among cells in terms of the functions they perform, for example, differences in secretion between secretory cells, like those of the intestinal epithelium and airways, and fibroblasts and astrocytes [11]. According to the specific case, these cisterns filled with protein products either have distinct destinations or may continue to reside in the ER. The main modifications, such as glycosylation, can mark the goal function of a protein, e.g., whether it must be secreted or form an integral part of the cell membrane. These proteins can undergo other modifications performed by the Golgi apparatus and form part of the secretory components and cell digestion, characterized by the vesicular traffic present in the cell, until they reach their destination. For these reasons, the ER is considered part of the secretory system, in which the Golgi apparatus participates, along with lysosomes and secretor vesicles as intermediary organelles of the process [2,12,13]. All these proteins are synthesized through the action of ribosomes in the part of the ER known as “rough” ER. The ER verifies the correct folding and formation of the tertiary structure of all recently synthesized proteins to ensure that they perform all their biological functions. This process, sometimes identified as proteostasis [14], is conducted by the ER with the support of resident proteins that perform chaperone functions, for example, GRP78/BiP and GRP94 [15]. Cells, however, can be affected by various types of stress that impact proteostasis. Studies have identified that the ER is a sensor organelle that detects intracellular modifications and alterations produced by changes in the extracellular space, such as the expression of mutant proteins, high secretory demands, loss of calcium homeostasis, altered lipid homeostasis, hypoxia, and viral infections like hepatitis B and C, among others [16,17,18]. These changes generate an increase in protein synthesis that sometimes exceeds the processing capacity of the ER, and misfolded proteins accumulate, increasing ER stress [16,18]. Various natural compounds, such as curcumin, can also induce stress responses in the ER [19,20]. These responses are utilized in various applications, especially therapeutic approaches.

## 4. Current Perspectives for the Treatment of Glioblastoma (GB)

### 4.1. Tumor Markers for the Treatment of GB

Research has established that in cases of aggressive cancers like GB, it is essential to identify and study the characteristics of the tumoral cells so that they can serve as markers when treatment begins. In recent years, the study of GRP78 has been emphasized due to this protein’s great potential for use as a marker in GB patients. At the same time, we now know that this protein plays crucial roles in GB proliferation, resistance, metastasis, and angiogenesis in hypoxic cellular environments characterized by glucose deprivation and inflammatory responses, among other factors that induce its expression. In other words, GRP78 can facilitate the adaptation of tumoral cells and promote their growth in an anoxic environment [21,22].

Multiple studies have proposed identifying GRP78 in the tumor microenvironment as a marker for diagnosing and characterizing this form of cancer, intending to improve diagnosis and prognosis in patients. In addition, treatment could also be based on molecular studies, not only on the histological analysis of biopsies. Mass spectrometry has also been used to show that GRP78 is expressed in large concentrations in tumoral cells; therefore, this justifies identifying it as a target for treating residual GB tumors. This approach is viable through the inhibition of GRP78, which permits a more integral form of management that facilitates the detection of the behavior and response of tumors [16]. In this regard, it is well documented that GRP78 is maintained in the ER in normal cells. However, in tumoral cells, in response to ER stress, it partially translocates to the cell surface (csGRP78), where it can form complexes with other proteins to mediate the transduction of the ER. csGRP78 in tumoral cells may be a viable target for specific therapies. A recent strategy using chimeric antigen receptor T (CAR-T) directed against csGRP78 resulted in the death of GB cells and GB stem cells in vitro and in vivo [23].

On the other hand, in a study that analyzed the databases The Cancer Genome Atlas (TCGA) and the Chinese Glioma Genome Atlas (CGGA) (689 samples included), seven genes were identified (*PDIA5*, *CASP8*, *CYBB*, *CCL2*, *HMOX1*, *ERN1*, *TP53*) to be associated with ER stress (ERS) in gliomas. This seven-gene signature could divide glioma patients into high-risk and low-risk subtypes. This ERS-related genetic signature is associated with a poor prognosis and higher malignancy in gliomas [24]. Ca^2+^ channels are abnormally expressed in various tumoral cells and are involved in human glioma progression. The calcium channel, voltage-dependent, T-type, alpha 1H subunit (*CACNA1H*) gene, which encodes T-type Ca^2+^ channel Cav3.2, functions as an oncogenic gene in glioma cells. Notably, *CACNA1H* knockdown decreased p-PERK, GRP78, CHOP, and ATF6 levels, indicating that *CACNA1H* knockdown activated ERS in U251 cells. The apoptosis of U251 cells was significantly increased after *CACNA1H* knockdown [25].

#### 4.1.1. Chemical and Pharmacological Approaches

Treating GB with chemotherapeutic agents has been the central focus in recent years. Most of these treatments center on triggering an ER stress response (ERSR) that activates the apoptotic pathway and the death of tumoral cells. Due to the nature and location of GBs, only some highly effective drugs have been developed. This is because numerous anti-GB medications either have a neurotoxic effect on patients, fail to cross the BBB, or attack the tumor inefficiently. As mentioned above, the search for new antitumor compounds that can successfully and directly induce an ERSR [22] ranges from ligands to oxygen-dependent metabolic routes in the cell [26] to the UPR of the ER [27].

Worldwide, the treatment indicated for most GB cases combines radiation and chemotherapy using the compound TMZ, a cytotoxic alkylating agent at the DNA level that inhibits cell division and the proliferation of tumoral cells. Most patients treated with TMZ present only a minimal improvement in survival (an average of just 2.5 months, with average survival of 14.6 vs. 12.1 months for TMZ-treated patients vs. patients under radiation treatment only, respectively) [28]. Due to these conditions, the past decade has brought the discovery and development of chemical compounds that effectively interact with TMZ to enhance its action in tumoral cells avoiding mechanisms such as resistance to TMZ treatment.

It is essential to understand that each chemotherapeutic scheme available today acts on a distinct mechanism in the tumoral cell. Compounds like chloroquine increase the cytotoxic effect of TMZ because they block GRP78-dependent autophagy, causing the ubiquitination of GRP78/BiP and inducing the expression of the proapoptotic protein CHOP/GADD-153 and poly (ADP-ribose) polymerase (PARP) proteins (among others). The result is an increase in the cytotoxicity of TMZ and apoptosis in vitro and in vivo [29]. Other descriptions show that SKIs (sphingosine kinase inhibitors) induce the accumulation of dihydrosphingosine and dihydroceramide, an increase in reactive oxygen species (ROS), an enhanced ERSR, and cell death regulated by caspase 3 [30].

The cell mechanisms of apoptosis and ERSR have also been shown to have greater effectiveness in treating GB cells. Compounds that have proven to have these effects will likely render the best results in glioma control. Radicol, a trinorguaiane-type sesquiterpene, demonstrated the induction of apoptosis via the ERS pathway by blocking v-akt murine thymoma viral oncogene homolog 1 (Akt)/mechanistic target of rapamycin kinase (mTOR) in TMZ-resistant tumoral cells [31], whereas fluoxetine, amiodarone, and desipramine act synergistically with TMZ to induce the CHOP-regulated apoptotic pathway [32,33,34]. Another mechanism that causes cell death is the accumulation of ROS in tumoral cells. For instance, salinomycin and tanshinone I induce the generation of ROS in GB cells, which results in apoptosis and autophagy in tumoral cells [26,35]. Luteolin, on the other hand, induces apoptosis in conjunction with mitochondrial dysfunction [36]. In contrast, dihydroartemisinin induces apoptosis and autophagy through the ER and mitochondria [37].

Because the ER is an organelle that also functions as a storage area for Ca^2+^ ions, in some cases, interrupting Ca^2+^ homeostasis can produce essential results in treating GB. In this strategy, sulindac-sulfide triggers mitochondrial release of Ca^2+^ and induces an ERSR and toxicity against GB cells [38]. A recent development consisted in applying combinations in which the synergy of quercetin and chloroquine caused caspase-dependent cell death by interrupting Ca^2+^ homeostasis and increasing autolysosomes and lysosomes via the increase in the load of undigested cell compounds, leading to cell death. At the same time, cytotoxicity is caused by mitochondrial and ER stress [39]. Phenethyl isothiocyanate (PEITC) is another apoptosis inducer in GB that releases ROS and Ca^2+^, inhibiting the growth of tumoral cells [40].

Other compounds, such as celecoxib (2,5-dimethyl celecoxib), act independently by generating an antiangiogenic effect in GB (reducing tumor size and microvasculature density) and suppressing the proliferation of tumor-associated brain endothelial cells [41] or by increasing sensitivity to radiotherapy to induce autophagy in hypoxic cells via ERSR overload [42]. The inhibition of protein disulfide isomerase (PDI) also demonstrated sensibilization to radiotherapy due to the accumulation of damaged DNA and the incapacity to repair it [27].

Certain aspects of the tumor microenvironment and its development must be considered to propose effective therapeutic approaches to cases of GB. One of the most critical aspects that must be directly tackled is the set of characteristics that give rise to metastasis, e.g., the epithelial–mesenchymal transition (EMT). It is also necessary to keep in mind that one of the most effective mechanisms for inducing the death of tumoral cells is paraptosis. Sinomenin hydrochloride (SH), an alkaloid derived from *Sinomenium acutum* Rehd. et Wils. (Fam. Menispermaceae), showed suppression of the expression of matrix metalloproteinases-2 and -9 (MMP-2/-9) and, as a result, metastasis suppression in GB cases. In addition, this compound arrests the cell cycle and inhibits the expression of NF-kB-p65 (an MMP-2/-9 promotor) while promoting an ERSR and autophagy by inducing inhibition of the EMT, inhibiting the phenotypic transition characteristic of metastasis [43]. The paraptosis cell death mechanism mentioned previously is much more effective because it is unrelated to apoptotic pathways, meaning that it overcomes the resistance of tumoral cells. It has been reported that luteolin improves death by paraptosis by inducing N-methyl-4-isoleucine cyclosporin (NIM811), that is, a cyclophilin anchor inhibitor that results in early activation of ERSRs, autophagy, and signaling via mTOR and the induction of paraptosis via significant vacuolization [44]. Finally, studies have demonstrated that depending on the concentration, curcumin can induce two types of cell death in glioma cell lines. At 100 mM, cells die by methuosis [45], whereas at 50 mM, they die by paraptosis [19]. In addition, the ERSR induces the degradation of many microRNAs in such a way that those that are increased foster cell death by activating at least two essential pathways: p53-Bcl-2 and insulin-AKT. The former is a pathway activated by reducing the expression of the gene, whereas the latter, involving the AKT protein, is a route of cell death in glioma cells associated with paraptosis [19].

#### 4.1.2. Derivatives of Natural Compounds in Treating GB

The resurgence of traditional medicine in the 21st century has also played an essential role in cancer research. Many natural compounds have shown effectiveness in treatments designed to control aggressive tumors. Significant advances have been made, mainly in compounds derived from plants, because researchers have discovered that some alkaloids—evodiamine, for example, derived from the dried fruit of *Evodia rutaecarpa—*trigger cell death via Ca^2+^-regulated autophagy and mitochondrial apoptosis [46].

Other compounds have demonstrated efficacy in inducing apoptosis in GB cells. These compounds do so through signaling via caspases 3, 7, 8, 9, and 12 [47,48], which are not induced by the UPR [49]. Among the derivatives that have shown favorable results, we can mention Wogonin, extracted from *Scutellaria baicalensis* Georgi, which induces ERSR and, as a result, apoptosis in tumor cells while generating large amounts of ROS [49]. Shikonin, extracted from the roots of *Lithospermum erythrorhizon*, reduces the viability of GB cells, causing death due to a cytotoxic effect [50]. Another critical advance was achieved with Asiatic acid (ASA), which reduces the viability of GB cells more effectively than TMZ because it triggers apoptosis regulated by the caspases mentioned above and the expression of the Bcl2 family, which has no toxic effects on nontumoral cells [47]. Expression of DNA damage-inducible transcript 3 (*DDIT3*) was achieved using a derivative of the herb Nan-Chai-Hu (*Bupleurum scorzonerifolium*), isochaihulactone (K8). This compound provides a regulatory effect by increasing the ERSR, the expression of *DDIT3*, and the non-steroidal anti-inflammatory drug-activated gene-1 (*NAG-1*)-regulated anti-inflammatory response, which, finally, leads to the activation of caspases 3 and 9, triggering apoptosis in GB cells [51]. Korov activates caspases 3, 8, and 9, triggering apoptosis mediated by mitochondria and the ERSR, a mechanism which phosphorylates PERK and eIF2α and increases IRE1α and the expression of XBP-1s while elevating GRP78, CHOP, and caspase anchor 12 [52].

A helpful cellular response produced by natural compounds is autophagy. One approach uses yessotoxin, a polycyclic ether extracted from dinoflagellates. This compound has been studied due to its effect on cell adhesion and proliferation in epithelial cells. It is essential to mention that the critical mechanism induced by yessotoxin is cell death by paraptosis because we know that this derivative alters the metabolism of lipids in glioma cells while also causing the UPR through the phosphorylation of PERK and EIF2α. It causes the arrest of cell proliferation through inhibition of protein synthesis, triggering autophagy in tumoral cells and, finally, achieving the inhibition of mTOR, the primary regulator of apoptotic processes [53]. Another compound that induces cell death by paraptosis is ophiobolin A, which interrupts the homeostasis of sulfhydryls, activating the ERSR and causing both CHOP expression and the appearance of apoptotic vacuoles in the ER [54].

As described in the previous section, one of the main obstacles in selecting a treatment for GB is the capacity of different compounds to cross the BBB. Some organic compounds, like piperlongumine, successfully cross this barrier while protecting nontumoral brain cells and directly activate mechanisms in tumoral cells by repressing the expression of PRDX4-like genes (where PRDX4 is an antioxidant protein), which generate an increase in ROS in gliomas that triggers ERSR and thus reduces the viability of GB cells [55].

Finally, it is important to note that these compounds can interact with chemotherapeutic treatment using TMZ. Previously described examples of this involved the use of bufothionine [56] and dihydrotansinone as inducers of ERSRs [57] or the use of epigallocatechin 3-gallate (EGCG), found in green tea, which has been shown to increase survival in murine study models by making GB cells more sensitive to TMZ by inhibiting GRP78 [28] (Table 1).

### 4.2. Beginning of Stress in the ER: ERAD and UPR Responses

The accumulation of large amounts of inadequately folded proteins due to any type of stress triggers the ER to act on such high misfolded protein load to recover proteostasis. These proteins are removed from the ER via a protein channel called dislocon [58], assisted by ATPase p97 activity (functional ATPase), and carried to the cytoplasm, where they are degraded through ubiquitylation in the 26S proteasome. This process is known as ER-associated degradation, or ERAD (Figure 1A) [12,13,59,60].

If the ERAD response does not resolve the problem of stress affecting the ER, then the unfolded protein response (UPR) is activated as a final measure. The UPR begins with the activation of three stress sensors in the ER. These are membrane proteins with domains both on the cytoplasmic side and in the lumen of the ER. They are important because they constitute the effector parts of each protein. What these three proteins have in common is that, generally, before the UPR, the domain found in the lumen of the ER constantly interacts with the master chaperone of the reticulum, glucose-regulated protein 78, (GRP78)/binding-immunoglobulin protein (BiP) to control the response (Figure 1A). During the UPR, on the other hand, the high levels of the chaperone that this ER response demands allow for the uncovering of these domains, which can simultaneously sense the presence of denaturalized proteins and then transmit the signal to its cytoplasmic domain, which performs diverse functions (Figure 1B). These sensor proteins are PERK (pancreatic endoplasmic reticulum kinase) [2,10,11,61], IRE1 (inositol-requiring enzyme 1), and ATF6 (activating transcription factor 6) [61,62]. The three-protein sensor system is highly conserved from yeasts to complex organisms like mammals, including humans [63].

Another of the UPR’s goals is to arrest the synthesis of some proteins, such as E2F transcription factor-1 (E2F1), p53 upregulated modulator of apoptosis (PUMA), phorbol-12-myristate-13-acetate-induced protein 1 (NOXA), to allow for cell survival [64] and the synthesis of other proteins that are the leading causes of stress. Another goal is allowing for the synthesis of transcription factors like Ying-Yang-1 (YY1) [65] to facilitate the transcription of chaperones (GRP78/BiP, for example), whose presence in the UPR is required. In addition, the UPR controls metabolism, so it stops the synthesis of proteins in various pathways that participate in the metabolism of lipids and glucose (Figure 1B) [13,63].

The importance of studying and understanding the UPR mechanism in tumoral cells resides in the fact that it can produce significant changes in cells characterized by a malignant phenotype, such as inducing angiogenesis, increasing the capacity for invasion and metastasis, causing metabolic modifications (anaerobic/Warburg effect), evading the immune response, and causing genetic instability, all of which favor the oncogenic process [16,66,67]. Some reports show that the high-grade tumors that characteristically reduce patient survival often show increased expression of GRP78/BiP [68].

This increase in GRP78/BiP protects glioma cells from ER stress responses [69], so when this expression is found, it has been shown to correlate with a poor prognosis in some patient studies [70]. It has been verified that GRP78/BiP regulates the growth of glioma cells. In this regard, the axis made up of ubiquitin-conjugating enzyme E2 T (UBE2T)/GRP78/epithelial–mesenchymal transition (EMT) is capable of favorably modulating the progression and recurrence of glioblastomas (Figure 1B) [68]. 

All these types of stress produce an uncontrolled increase in protein synthesis that, in most cases, appears as a morphological alteration characterized by the accumulation of variously sized vesicles. Sometimes, these fuse to form clear cytoplasm occupied by large vesicles that, according to detailed studies and analyses, correspond to dilatations, primarily of the ER. Some of these cells have the appearance of a “scrambled egg”, an alteration called paraptosis that depends on protein synthesis (Figure 2A); this process is reversible in the presence of cycloheximide, a protein synthesis inhibitor, and the cell can recover its normal phenotype [19].

#### 4.2.1. PERK

PERK is a transmembrane protein expressed in nearly all tissues, especially in the pancreas, prostate, stomach, and thyroid. It is only found in the ER, where it resides as a monomer in the domain of the lumen, forming a protein complex with GRP78/BiP. By participating in the UPR, this protein controls multiple functions in the development of pluricellular organisms, including humans [71]. Its activity as a serine/threonine kinase has been described in rat pancreatic islets. It can phosphorylate the factor known as eukaryotic translation initiation factor 2 (eIF2a) to control and arrest the translational activity of messenger RNAs, in general, inside the ER [72]. The UPR permits the interruption of its interaction with GRP78/BiP. In this way, the dimerization of PERK and its autophosphorylation are performed. Phosphorylated PERK is activated and capable of phosphorylating eIF2a, which participates in the onset of protein synthesis in the ribosome. However, once phosphorylated, it loses this property and stops protein synthesis in the ER [73]. The group of proteins expressed includes activating transcription factor 4 (ATF4), whose function is to induce the expression of the C/EBP homologous protein, or CHOP [74]. These two proteins act as transcription factors that strengthen the UPR. CHOP does so by regulating anti- and proapoptotic genes, such as B-cell lymphoma 2 (BCL2), growth arrest and DNA damage-inducible gene 34 (GADD34), Tribbles pseudokinase 3 (TRB-3), and the double C2-like domain-containing protein alpha (DOC) family [75]. At the same time, ATF4 increases the expression of the genes that participate in importing amino acids, glutathione biosynthesis, and in general, the antioxidant response [76]. PERK also phosphorylates the protein nuclear factor erythroid 2-related factor 2 (NRF2), which, when modified, activates the antioxidant pathway that depends on it [77]. The control of protein synthesis also produces a reduction in the inhibitor of nuclear factor of kappa light polypeptide gene enhancer in B-cells 1 (NF-kB) (IkB), which increases the levels of the NF-kB protein that is translocated to the cell nucleus, where it acts as a transcription factor with the ability to increase the expression of the genes of various proinflammatory cytokines, such as IL-6 and IL-8, causing inflammation [78]. Finally, the sustained expression of the CHOP and ATF4 proteins synergically activates the stop genes in the cell cycle, activating cell death (Figure 2B) [79].

#### 4.2.2. IRE-1

It is essential to mention the existence of two genes that codify the two forms of IRE1, *ERN1* and *ERN2* (endoplasmic reticulum to nucleus signaling 1 and 2), which encode the IRE1α and IRE1β proteins, respectively. IRE1 responses always refer to the former because it is expressed ubiquitously in the body [2]. Like the PERK protein, this sensor protein has a domain in the lumen of the ER and is found to form a complex protein with GRP78/BiP under stress-free conditions or during proteostasis. However, UPR response activation, because of the activity of the chaperones induced by the excess of poorly folded proteins, causes the release of this domain of the IRE1 protein, allowing for the activation of the serine/threonine kinase domain, which performs the autophosphorylation and dimerization of IRE1 proteins in the residues Ser724, Ser726, and Ser729 [80] and activates their function as RNases. This is how the degradation of a broad range of RNAs (RNAm, microRNAs, RNAr), called response IRE1-dependent decay, or RIDD, starts [81]. The IRE1 protein also interacts with TRAF2 to trigger and facilitate apoptosis by recruiting and activating apoptosis signal-regulated kinase 1 (ASK1) and c-Jun NH2-terminal kinase (JNK) [82]. This activity of the RNase also permits the splicing of an intron of the mRNA of XBP1 (X-box protein1), which generates the XBP1 protein, which is the spliced form that functions as an essential factor in regulating the expression of genes directly related to the ER, whether through their expansion or exportation, or the degradation of poorly folded proteins [83,84]. This function of the XBP1 transcription factor regulates various metabolic pathways related to the biosynthesis of lipids, glucose metabolism, and the insulin signaling pathway.

Moreover, the IRE1 protein plays a role in the production of oxidative stress, which is related to cell survival and DNA reparation. Other functions regulated by this protein include cell differentiation and development due to XBP1′s ability to physically interact with diverse transcription factors, especially activating protein-1 (AP-1), activating transcription factor 6 (ATF6), hypoxic inductor factors a (HIF1a), and estrogen receptors [2]. The unspliced form, XBP1u, inhibits the transcription of XBP1s [85]. The activity of the RNase of IRE1 also regulates the degradation of its own mRNA as an autoregulating mechanism [86]. Finally, it is essential to mention that the IRE1 b form induces the translational repression of messengers by degrading the RNA 28S subunit, which affects the formation and assembly of the two subunits of ribosomes (Figure 2B) [87].

#### 4.2.3. ATF6

This protein of the ER, like the two other sensor proteins, PERK and IRE1, resides as an integral protein with a domain in the lumen of the ER and is made up of a protein complex with the protein GRP78/BiP in proteostasis. During the UPR, GRP78/BiP ceases to form a complex, allowing ATF6 to be activated. It is important to note that humans have two genes, ATF6A and ATF6B, that codify for the proteins ATF6α and ATF6β, respectively [88,89]. After activation, the ATF6α protein is exported from the ER to the Golgi apparatus, where it undergoes a maturation process that consists of a proteolytic cut that eliminates a C-terminal end, thus releasing an active 50 kDa protein called ATF6αf, which contains various domains, including that of the transcriptional activator, the domain for DNA binding, and its nuclear localization signal. This always refers to ATF6α because the β form lacks a transactivation domain [90,91]. With its transit signal in the nucleus, ATF6f translocates there, forming a heterodimer with the nuclear transcription factor Y subunit alpha (NF-Y) factor. Together, these interact with the ER (SEER) stress elements to increase transcription and the GRP78/BiP, CHOP, and XBP1 proteins supporting the UPR. It is a point of convergence of the three sensors in this pathway [92,93]. ATF6f also modulates the expression of other genes by forming heterodimers with YY-1 [94,95], CREB [96], cAMP responsive element binding protein 3 Like 3 (CREB3L3) [97], and transcription factor 1 (TF1) [98]. Finally, ATF6 can interact with XBP1 to form a heterodimer and activate some genes in the ERAD system described above. One of these is a mannosidase-like protein 1, EDEM1, which maintains ER homeostasis by processing poorly folded proteins and quickly sending them to be marked with ubiquitin and degraded in the ERAD system via autophagy when either dislocation or proteasomal degradation are impaired (Figure 2B, black color) [92,99].

## 5. Conclusions

The ER is a sensor organelle that verifies the correct folding and formation of the tertiary structure of all newly synthesized proteins to ensure that they perform all their biological functions intra- and extracellularly. In several diseases, including cancer, this mechanism is disrupted, and unfolded proteins accumulate in the ER. To restore homeostasis, the UPR mechanism is activated, with which cells develop adaptive programs; otherwise, apoptosis mechanisms are induced. Neoplastic cells use the UPR to overcome stress conditions and maintain cell viability, hence the importance of considering it a mechanism of therapeutic intervention in tumor progression. Future perspectives on the treatment of gliomas must be focused on processes that induce ERSRs, apoptosis, autophagy, and paraptosis to achieve tumoral cell death with greater efficacy when using antitumor proteins like GRP78/BiP and CHOP in glioma. GRP78 can facilitate the adaptation of tumoral cells and promote their growth in an anoxic environment. The search for new antitumor compounds that can successfully and directly induce an ERSR ranges from ligands to oxygen-dependent metabolic routes in the cell to the UPR of the ER and is impacting the biomedical area. In the treatment of GB, radiation is combined with chemotherapy using the compound TMZ, achieving a minimal improvement in survival. For this reason, the past decade has brought the discovery and development of chemical compounds and natural derivatives that effectively interact with TMZ to enhance its action in tumoral cells, either by inducing the CHOP-regulated apoptotic pathway or by making GB cells more sensitive to TMZ by inhibiting GRP78.

Finally, the primary biomedical studies on the responses of the ER and the UPR represent a solid base that could open new horizons in the treatment of gliomas because the UPR could activate new, undocumented cellular responses such as cellular dormancy, a critical response involved in cell survival. We must not lose sight of the fact that the selectivity of most pharmacological compounds or inhibitors of the UPR pathway can also affect normal cells and lead to a more severe risk, so more extensive studies on UPR modifiers are necessary.

## Figures and Tables

**Figure 1 ijms-25-04078-f001:**
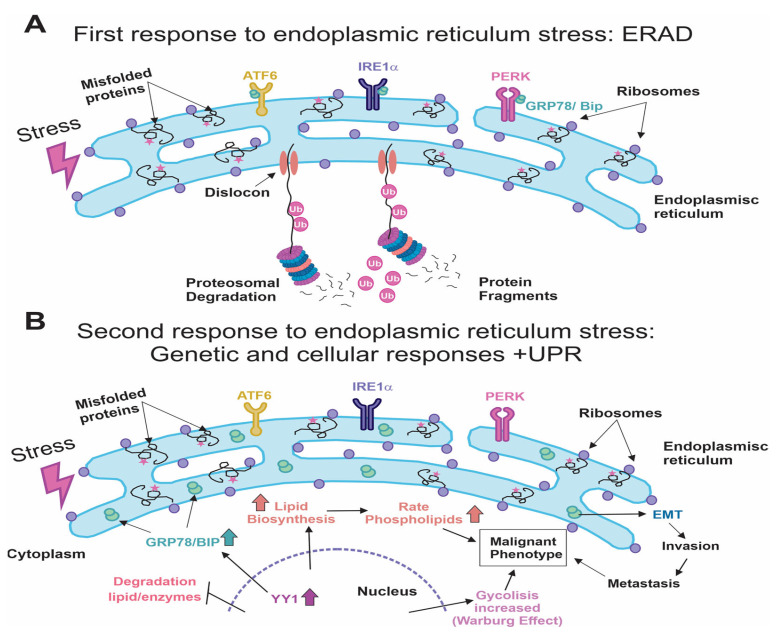
Initial response to ER stress. (**A**) The first response to stress is performed by ER-associated degradation, or ERAD: proteins inadequately folded proteins are translocated to cytoplasm via dislocon, and will be degraded by ubiquitination (Ub, pink circle) through the 26S proteasome. All domains of each protein sensor (PERK, IRE1α, and ATF6) are complexed with the main chaperone GRP78/BiP. (**B**) The unfolded protein response (UPR) is activated as a final response when stress is not resolved. The UPR begins with the activation of three stress sensors in the ER (PERK, IRE1α, and ATF6), and all leave the interaction with GRP78/BiP that is required due to an increase in inadequately folded proteins in ER lumen. High GRP78/BiP activates the mechanism epithelial mesenchymal transition (EMT) that is related to increased cell mobility and metastasis. There are increases of glycolysis and lipid biosynthesis and high turnover of phospholipids that are all related to the malignant phenotype.

**Figure 2 ijms-25-04078-f002:**
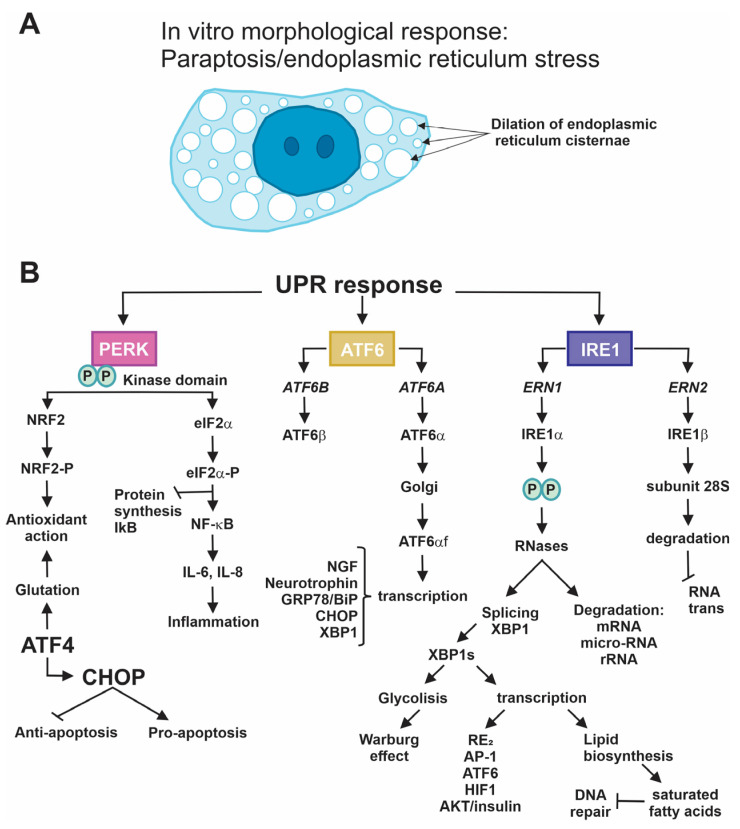
Main characteristics of the UPR response. (**A**) The cell phenotype of ER stressed, normally is the appearance of the “scrambled egg” that correspond to paraptosis, a classical look of the ER cisternae dilated. (**B**) Details of the UPR response given by the three RE stress sensors (PERK, ATF6, IRE1). PERK phosphorylates (P, green circle) NRF2 and eIF2α. In the first case, the antioxidant pathway is activated, and in the second case, protein synthesis is blocked. The ATF6 pathway is formed by the *ATF6A* and *ATF6B* genes that encode the ATF6α and β proteins, respectively. After its activation, the ATF6α protein is exported to the Golgi apparatus, where it undergoes a maturation process, releasing an active protein called ATF6αf, with a DNA binding domain. ATF6αf translocates to the nucleus, where it increases transcription of the proteins GRP78/BiP, CHOP, and XBP1, among others. ATF6 can interact with XBP1 to form a heterodimer and activate some genes of the ERAD system. Finally, persistent and unresolved ER stress leads to autophosphorylation (P, green circle) of IRE1α which increases its function as RNase and degrades a wide range of RNAs. It also generates the splicing of XBP1, whose function is to regulate the expression of genes directly related to the ER. The IRE1α protein also interacts with TRAF2 to activate and facilitate apoptosis. CHOP and XBP1 show the convergence of the three pathways of stress response.

**Table 1 ijms-25-04078-t001:** Chemical pharmaceutical compounds and natural derivatives with therapeutical effects in glioma cells.

Compound	Effects	Ref
Temozolomide (TMZ)	Citotoxic, DNA-alkylating. Proliferation inhibitor.	[28]
Chloroquine	Increases the cytotoxicity of TMZ. Ubiquitination of GRP78, CHOP, PARP. Inhibition of apoptosis regulated by GRP78. Promotes ERS.	[29]
Sphingosine kinase inhibitors (SKIs)	Accumulation of dhSph and dhCer, ROS. ERS induction. Cell death regulated by caspase 3.	[30]
Radicol	Apoptosis induction via ERS. Akt/mTOR blockade in TMZ-resistant tumor cells.	[31]
Fluoxetine Desipramine Amiodarone	Synergy with TMZ. Induction of apoptosis by CHOP.	[32,33]
Salinomycin Tanshinone I	Generation of ROS. Apoptosis and autophagy of tumor cells.	[35,43]
Luteolin	Mitochondrial dysfunction triggered by apoptosis.	[36]
Dihydroartemisinin	Induction of apoptosis and autophagy by ER and mitochondria.	[37]
Sulindac-Sulfide	Mitochondrial release of Ca^2+^ ions. ERSR induction. Cellular toxicity.	[37]
Quercetin and Chloroquine	Disruption of Ca^2+^ homeostasis. Induction of caspase-dependent cell death. Increase in autolysosomes and lysosomes. Cell death due to mitochondrial stress and ER.	[39]
Phenethyl isothiocyanate (PEITC)	Induction of apoptosis by release of ROS and Ca^2+^. Inhibits tumor cell growth.	[40]
Celecoxib (2,5-Dimetil Celecoxib)	Antiangiogenic effect. Reduces tumor size and microvessel density. Suppresses proliferation of tumor endothelial cells. Increases sensitivity of radiotherapy.	[41]
Sinomenine hydrochloride (SH)	Suppression of MMP-2/-9 expression. Reduces metastasis. Stops cell cycle. Inhibits expression of NFkB-p65. Promotes ERSR and autophagy. EMT inhibition.	[43]
Luteolin	Cell death due to paraptosis and vacuolization. Induction of NIM811. Inhibits cyclophilin anchoring. Early activation of ERSR. Autophagy. mTOR signaling.	[44]
Curcumin	100 µM: death from methuosis. 50 µM: death by paraptosis. Induces ERSR, resulting in microRNA degradation. Activation of the p53-Bcl-2 and insulin-AKT pathways. AKT gene expression decreases.	[45]
**Deri. Nat.** **Compounds**	**Effects**	**Ref**
Evodiamine	Derived from the dried fruit of Evodia rutaecarpa. Ca^2+^-regulated autophagy and mitochondrial apoptosis.	[46]
Asiatic acid (AsA)	Reduced viability of GB cells. Expression of Bcl2 without cytotoxic effects for nontumor cells.	[47]
Wogonin	Derived from *Scutellaria baicalensis* Georgi. ERSR induction; tumor cell apoptosis. ROS elevation.	[49]
Shikonin	Derived from the roots of Lithospermum erythrorhizon. Cytotoxic effect; reduced cellular sensitivity. Caspase-regulated apoptosis.	[50]
Isochaihulactone (K8)	Extracted from Bupleurum scorzonerifolium. Increases ERSR; DDIT3 expression. NAG-1-mediated inflammatory response. Activation of caspases 3 and 9; apoptosis.	[51]
DAW22	Coumarin sesquiterpene derived from Ferula ferulaeoides (Steud) Korov. Activation of caspases 3, 8, 9 and 12; mitochondria-mediated apoptosis and ERSR. Expression of XBP-1, elevation of GRP78 and CHOP. PERK and eIF2α phosphorylation, IRE1α increase.	[52]
Yessotoxin	Polycyclic ether extracted from dinoflagellates. Cell death due to paraptosis. Altered lipid metabolism. UPR induction; PERK and EIF2α phosphorylation. mTOR inhibition.	[53]
Ophiobolin A	Disrupts sulfhydryl homeostasis. ERSR activation; CHOP induction. Apoptotic vacuoles in ER.	[54]
Piperlongumine	Repression of PRDX4 expression. Increased ROS. ERSR activation.	[55]
Bufothionine Dihydrotanshinone Epigallocatechin 3-gallate	Use in synergy with TMZ. Increased chemosensitivity to TMZ. GRP78 inhibition. Survival in murine models.	[28,56,57]

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
