# Peer review of "Endoplasmic Reticulum Stress in Gliomas: Exploiting a Dual-Effect Dysfunction through Chemical Pharmaceutical Compounds and Natural Derivatives for Therapeutical Uses"

_ijms, 2024, doi:10.3390/ijms25074078_

Round 1

Reviewer 1 Report

Comments and Suggestions for Authors

Reviewer comments and suggestions

The authors in this study discussed the unfolded protein response (UPR) and endoplasmic response-associated degradation (ERAD) about gliomas and suggested that it can generate two responses: cell survival and cell death by different mechanisms. The UPR stress response can control and kill glioma cells, and moreover, the authors suggested that it can be advantageous from this dysfunction with therapeutic advantages using chemical, pharmaceutical compounds and natural derivatives.

Overall, the manuscript was not well written. And it should be thoroughly revised and need English correction from a native speaker. Additionally, a few concerns or comments are required to be explained or modified.

  1. Line 24-25 Sentence needs to be written
  2. I think the abstract should be modified based on the text content in the manuscript
  3. In the introduction section, at the end, the authors need to explain the study content that is presented afterward.
  4. Line 76-77 Please explain the sentence in a better and more understanding way
  5. Line 115-116 studies were used by the authors but did not add appropriate references for the same
  6. Line 118-119 These points should be well discussed
  7. Line 124 first time used this word so it should be in full form (GB)
  8. is this was a correct line “Apoptosis of U251 cells was significantly improved after CACNA1H knockdown [64].”
  9. Line 162 GBM AND ERSR first time used so it should be in full form
  10. Line 181 inhibition of apoptosis could lead cytotoxicity.
  11. Line 256 (DDIT3) should be in full form
  12. Comments for figure 1 I think the authors could prepare a more appropriate and professional diagram
  13. Comments for Figure 2 Legend should be described
  14. Line 395 It needed the same representation of alpha 
  15.      and beta in the figure as well as in text
  16. Line 444 -445 The authors need to mention it protective effects or apoptotic effects.
  17. Please check references 21, 76

Author Response

8th January 2024

Prof. Dr. Maurizio Battino

Editor-in -Chief IJMS

Department of Odontostomatologic and Specialized Clinical Sciences,

Sez-Biochimica, Faculty of Medicine, Università Politecnica delle Marche, Ancona, Italy

We present for the second time the manuscript entitled: “Endoplasmic reticulum stress in gliomas: a dysfunction with two effects taken advantage of by chemical, pharmaceutical compounds and natural derivatives with therapeutical use” by Daniel García-López, et. al. We appreciate the reviewers' comments and suggestions.

  1. REFEREE: Overall, the manuscript was not well written. And it should be thoroughly revised and need English correction from a native speaker.

RESPONSE: The manuscript was sent to MDPI's English Language Editing Services.

  1. REFEREE: Line 24-25 Sentence needs to be written.

RESPONSE: Correction in paragraphs 23-28.

  1. REFEREE: Comments I think the abstract should be modified based on the text content in the manuscript.

RESPONSE: The information in the abstract was redrafted to reflect the most important points of the manuscript. In addition, a statement was added that clarifies and completes the information. Correction in paragraphs 23-28.

  1. REFEREE: In the introduction section, at the end, the authors need to explain the study content that is presented afterward.

RESPONSE: At the end of the introduction, a paragraph was added explaining the content of the study. Corrected in paragraphs 41-46.

  1. REFEREE: Line 76-77 Please explain the sentence in a better and more understanding way:

RESPONSE: Information on the prognosis of patients with IDH1 mutation was written more clearly. Corrected in paragraphs 85-96.

  1. REFEREE: Line 115-116 studies were used by the authors but did not add appropriate references for the same.

RESPONSE: The references corresponding to paragraphs 132 to 138 were appropriately added and the information was expanded so that it is complete and clear. In these paragraphs, the document cites Peñaranda et al., Markoulis et al, and Hetz et al. Line 132-138

  1. REFEREE: Line 118-119 These points should be well discussed.

RESPONSE: The information was modified, correction in paragraphs 134 to 136.

  1. REFEREE: Line 124 first time used this word so it should be in full form (GB):

RESPONSE: The meaning of the abbreviation was placed. Corrected line 141.

  1. REFEREE: Is this was a correct line “Apoptosis of U251 cells was significantly improved after CACNA1H knockdown [64].

RESPONSE: The statement was corrected as follows: “The apoptosis of U251 cells was significantly increased after CACNA1H knockdown”. Corrected lines 176 and 177.

  1. REFEREE: Line 162 GBM and ERSR first time used so it should be in full form.

RESPONSE: GBM was removed and changed to GB. Corrected line 179.

The meaning of ERSR was placed. Corrected line 180.

  1. REFEREE: Line 181 inhibition of apoptosis could lead cytotoxicity.

RESPONSE: It was corrected by the statement: “The result is an increase in the cytotoxicity of TMZ and apoptosis in vitro and in vivo”. Corrected line 201.

  1. REFEREE: Line 256 (DDIT3) should be in full form.

RESPONSE: Corrected line 276.

  1. REFEREE: Comments for figure 1 I think the authors could prepare a more appropriate and professional diagram.

RESPONSE: The two figures were designed in a more professional way. Corrected line 334 and 405, respectively.

  1. REFEREE: Comments for Figure 2 Legend should be described.

RESPONSE: Corrected line 408-417.

  1. REFEREE: Line 395 It needed the same representation of alpha  and beta in the figure as well as in text.

RESPONSE: Corrected in foot figure 2: line 408 and in the text line 422.

  1. REFEREE: Line 444 -445 The authors need to mention it protective effects or apoptotic effects.

RESPONSE: Corrected line 473.

  1. REFEREE: Please check references 21, 76.

RESPONSE: The two references were revised, reference 21 is now 61 and reference 76 is now 37, this is because the information was moved, and we did not notice the error since the initial submission to the journal. In general, references were reviewed and ordered according to their appearance in the text.

Reviewer 2 Report

Comments and Suggestions for Authors

Manuscript number: ijms-2722002 Manuscript type: Review, titled "Endoplasmic reticulum stress in gliomas: a dual-acting dysfunction exploited by chemical, pharmaceutical and natural derivatives with therapeutic applications"

The article discusses the issue of GB cases in both children and adults, i.e. the survival time of both categories of patients has not changed for over 20 years. Despite a more complete understanding of the molecular mechanisms involved in the development of these types of cancers, which most often appear in the CNS. The authors described in detail two processes: ERAD and UPR, which may regulate the emergence of the malignant phenotype, triggering EMT, invasion and metastasis. This manuscript defines a serious problem and points to its enormous complexity, but also proposes the use of standard chemotherapy enriched with natural derivatives. 

The citation used in the text are very current and very right on.

Not very important: unexpanded abbreviations GB, GBM. 

Also, ERS (page 8 line151) is explained and line159 the same page.

Moreover, I would develop solid conclusions or discussions. Out of 15 pages of the manuscript, not quite seven lines of conclusions. 

And further it is not this type of manuscript; we could say: It's easy to get through this (?).

Comments on the Quality of English Language

And father it is not this type of manuscript; we could say: It's easy to get through this. Particularly from the page 3, 4:. Current Perspectives FOPR THE Treatment of CB.

Author Response

8th January 2024

Prof. Dr. Maurizio Battino

Editor-in -Chief IJMS

Department of Odontostomatologic and Specialized Clinical Sciences,

Sez-Biochimica, Faculty of Medicine, Università Politecnica delle Marche, Ancona, Italy

We present for the second time the manuscript entitled: “Endoplasmic reticulum stress in gliomas: a dysfunction with two effects taken advantage of by chemical, pharmaceutical compounds and natural derivatives with therapeutical use” by Daniel García-López, et. al. We appreciate the reviewers' comments and suggestions.

REFEREE 2. We appreciate your interest in reviewing this manuscript and appreciate your comments and suggestions. Below we mention the changes made indicating the line number and highlighted in red in the text.

  1. REFEREE: Not very important: unexpanded abbreviations GB, GBM.

RESPONSE:  Added meaning and abbreviation of GB. Corrected line 141.

  1. REFEREE: Also, ERS (page 8 line151) is explained and line159 the same page.

RESPONSE: Corrected line 169.

  1. REFEREE: Moreover, I would develop solid conclusions or discussions. Out of 15 pages of the manuscript, not quite seven lines of conclusions. 

RESPONSE: The conclusions were expanded in such a way that they contain the important points of the manuscript. Corrected lines 475-502.

Round 2

Reviewer 1 Report

Comments and Suggestions for Authors

All comments has been addressed, no more comments. Thank you